# PROGRESSIVE STAGE-AWARE REINFORCEMENT FOR FINE-TUNING VISION-LANGUAGE-ACTION MODELS

## ABSTRACT

Recent advances in Vision-Language-Action (VLA) models, powered by large language models and reinforcement learning-based fine-tuning, have shown remarkable progress in robotic manipulation. Existing methods often treat long-horizon actions as linguistic sequences and apply trajectory-level optimization methods such as Trajectory-wise Preference Optimization (TPO) or Proximal Policy Optimization (PPO), leading to coarse credit assignment and unstable training. However, unlike language, where a unified semantic meaning is preserved despite flexible sentence order, action trajectories progress through causally chained stages with different learning difficulties. This motivates progressive stage optimization. Thereby, we present **St**age-**A**ware **Re**inforcement (STARE), a module that decomposes a long-horizon action trajectory into semantically meaningful stages and provides dense, interpretable, and stage-aligned reinforcement signals. Integrating STARE into TPO and PPO, we yield Stage-Aware TPO (STA-TPO) and Stage-Aware PPO (STA-PPO) for offline stage-wise preference and online intra-stage interaction, respectively. Further building on supervised fine-tuning as initialization, we propose the **I**mitation→**P**reference→**I**nteraction (IPI), a serial fine-tuning pipeline for improving action accuracy in VLA models. Experiments on SimplerEnv and ManiSkill3 demonstrate substantial gains, achieving state-of-the-art success rates of 98.0% on SimplerEnv and 96.4% on ManiSkill3 tasks. Our code will be released publicly.

## 1 INTRODUCTION

Large-scale Vision–Language–Action (VLA) models (Zitkovich et al., 2023; Ghosh et al., 2024; Kim et al., 2024; Black et al., 2024; 2025) have recently emerged as powerful generalist policies for robotic manipulation. These models unify image, language, and action modalities within a single architecture, enabling robots to interpret multimodal inputs and generate executable actions. Pretrained on massive-scale multimodal data (O'Neill et al., 2024; Walke et al., 2023), VLA models provide strong priors that can be efficiently adapted to diverse downstream tasks through fine-tuning, avoiding the need for retraining from scratch.

Recent development of large-scale VLA models has been rapidly driven by the success of vision–language models (VLMs) and large language models (LLMs), as their output, i.e., action trajectories and sentences, can both be represented as sequential data (Zitkovich et al., 2023; O'Neill et al., 2024; Ghosh et al., 2024). Consequently, many developed fine-tuning techniques, such as supervised fine-tuning (SFT), reinforcement learning from feedback (RLFT) (Ouyang et al., 2022), direct preference optimization (DPO) (Rafailov et al., 2023), and Proximal Policy Optimization (PPO) (Schulman et al., 2017), have been straightforwardly adopted for VLA models . However, directly applying these methods to fine-tune on the whole action trajectories remains cumbersome and often inefficient, as the large optimization space makes credit assignment across long-horizon trajectories highly ambiguous. Unlike language reasoning, where optimization depends on a holistic understanding of sentences without strict ordering, an action trajectory naturally decomposes into semantically distinct stages that are causally chained and vary in difficulty. For example, as a pick-and-place task illustrated in Figure 1, *Reach* must precede *Grasp*, which in turn precedes *Transport* and *Place*. *Reach* and *Transport* are relatively easy with simple optimization objectives, while *Grasp* and *Place* are more challenging as they require precise geometric constraints. Overall task success hinges on correct progression through all stages. This fundamental characteristic moti-

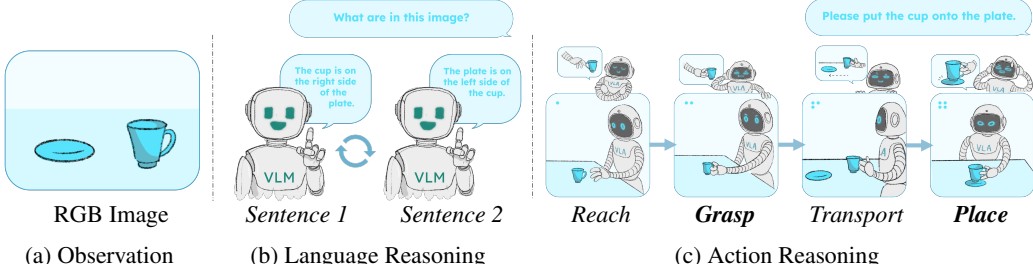

| RGB Image | Sentence 1 | Sentence 2 | Reach | **Grasp** | Transport | **Place** |
|-----------|-----------|-----------|-------|-----------|-----------|-----------|
| (a) Observation | (b) Language Reasoning | | (c) Action Reasoning | | | |

Figure 1: **Language Reasoning vs. Action Reasoning.** Given an RGB image as the observation (a), the language model (b) is asked to describe the content in the image, and produces *Sentence 1* and *Sentence 2*. These sentences are flexibly ordered and jointly contribute to the global meaning required to answer the question. In contrast, the VLA model (c), when instructed to place the cup onto the plate, generates an action trajectory composed of semantically meaningful stages (*Reach→Grasp→Transport→Place*), which follow a strict order and vary in difficulty (with the more challenging stages shown in bold).

vates stage-aware objectives rather than monolithic trajectory-level optimization, which remains the predominant paradigm in current VLA fine-tuning.

In this paper, we design *Stage-Aware **Re**inforcement* (STARE), a plug-in module that decomposes action trajectories into progressive stages with dense reward signals based on task-specific semantics. Given a trajectory, either in the collected data or during the model's rollout, STARE employs a stage separator to identify *when* stage transitions occur, based on the translation and orientation of an end-effector. A stage calculator computes a stage cost and per-step rewards to evaluate *how well* each stage is executed. In this way, STARE not only annotates stage-wise actions but also assesses partial successes and failures within a trajectory. We leverage STARE for offline fine-tuning via *Stage-Aware Trajectory-Wise Preference Optimization* (STA-TPO), which constructs pairwise preferences at the stage level. By incorporating stage costs, STA-TPO propagates precise gradient signals to specific action stages, enabling progressive learning and credit assignment not only between success and failure but also among varying degrees of success. For online fine-tuning, we introduce STARE to *Stage-Aware Proximal Policy Optimization* (STA-PPO), which reshapes sparse terminal rewards to dense interaction rewards. By providing this progressive feedback, STA-PPO stabilizes intra-stage updates, especially for complex manipulation tasks that require dense guidance. Conceptually, STA-TPO and STA-PPO are reminiscent of curriculum learning (Bengio et al., 2009), where training is organized along an ordered sequence of subtasks to ease optimization and improve generalization. However, unlike conventional curricula that progress strictly from easy to hard, our stage-aware design enforces semantic continuity across stages, ensuring that optimization respects causal dependencies in stages.

To sufficiently fine-tune a pre-trained VLA model with STA-TPO and STA-PPO, we integrate these two algorithms with SFT as an initialization into a serial tri-step fine-tuning pipeline, *Imitation→Preference→Interaction* (IPI). The IPI framework first finetunes a VLA model from expert demonstrations via SFT, then further optimizes it according to offline stage-aware preferences using STA-TPO, and finally refines it based on stage-aware interaction in online environments using STA-PPO. In contrast to existing VLA fine-tuning strategies, IPI offers two key advantages: first, it explicitly models the multi-stage structure of robot trajectories, enabling more precise credit assignment rather than monolithic trajectory-level optimization (Zhang et al., 2024). Second, compared to other methods that treat offline and online fine-tuning as disjoint processes (Zhang et al., 2024; Lu et al., 2025a; Chen et al., 2025c), IPI unifies them under a single framework, enabling stage-wise preference alignment and intra-stage interactions. Extensive experiments show that IPI not only improves in-distribution success rates but also substantially enhances out-of-distribution generalization, underscoring the importance of multi-stage reward design in VLA fine-tuning.

Our contributions are summarized as: **(i)** We design STARE, a rule-based module that decomposes trajectories into semantically meaningful stages, enabling fine-grained supervision beyond trajectory-level signals. **(ii)** Based on (i), we propose stage-aware fine-tuning methods: STA-TPO for offline stage-wise preference alignment and STA-PPO for online intra-stage interaction, both providing more precise credit assignment and improved sample efficiency. **(iii)** We unify supervised fine-tuning, STA-TPO, and STA-PPO into IPI, a serial tri-step pipeline for fine-tuning VLA models, and validate it on the benchmarking frameworks SimplerEnv and ManiSkill3, showing that IPI achieves state-of-the-art success rates and substantially improves out-of-distribution generalization.

## 2 RELATED WORK

**Long-horizon Robotic Manipulation Tasks in RL** Long-horizon robotic manipulation involves completing a sequence of sub-tasks with frequent state and environment changes. Applying RL to such tasks is challenging due to sparse rewards, credit assignment, error accumulation, and high-dimensional state spaces. To address these issues, Plan-Seq-Learn (Dalal et al., 2024) leverages language models for high-level planning and RL for low-level control, enabling end-to-end execution from visual input to complex tasks. $AC^3$ (Yang et al., 2025) learns continuous action chunks with intrinsic rewards to mitigate sparsity. $DEMO^3$ (Escoriza et al., 2025) augments limited demonstrations with a world model and stage-wise dense rewards to improve sample efficiency. Robo-Horizon (Chen et al., 2025d) employs LLMs to generate sub-goals and rewards, integrated with multi-view world models and planning to achieve high success rates. ARCH (Sun et al., 2025) combines high-level policy selection with a primitive skill library to tackle contact-rich assembly. SARM (Chen et al., 2025b) uses subtask-annotated reward modeling to filter demonstration quality, enabling robust long-horizon deformable object manipulation. Building on these inspirations that divide goals into sub-goals, we take a further step with stage-aware reinforcement, decomposing trajectories into semantically meaningful stages. This provides denser feedback and enables progressive optimization, making RL more effective for long-horizon VLA tasks.

**RL Fine-tuning for LLMs** RL fine-tuning is a widely used approach for aligning LLMs. The most prominent method is Reinforcement Learning from Human Feedback (RLHF) (Ouyang et al., 2022), where a reward model trained from human preference data guides algorithms such as policy gradient or PPO to align outputs with human expectations. While highly successful, RLHF suffers from costly data collection and unstable training. To mitigate these issues, DPO (Rafailov et al., 2023) eliminates the reward model by directly optimizing on preference comparisons, simplifying training and improving stability. Further variants such as RLAIF (Lee et al., 2023) and RAFT (Dong et al., 2023) refine the framework. DeepSeek-R1 (Guo et al., 2025a) employs GRPO, which samples multiple responses per prompt and uses their relative performance within the group to compute advantages. As a subclass of LLMs, Large Reasoning Models (LRMs) (Zhang et al., 2025b) utilize Chain-of-Thought (CoT) (Wei et al., 2022) or Process Reward Model (PRM) (Lightman et al., 2023) for multi-step reasoning and face challenges akin to long-horizon VLA tasks, including sparse rewards and difficult credit assignment. This motivates our stage-aware reinforcement approach for VLA models.

**RL Fine-tuning for VLAs** Recent studies explore RL as a fine-tuning paradigm for VLA models. GRAPE (Zhang et al., 2024) adapts DPO (Rafailov et al., 2023) to trajectory-level preferences to propose TPO, while ConRFT (Chen et al., 2025c) alternates RL and SFT in real-world settings. ReinboT (Zhang et al., 2025a) designs dense rewards, and Guo et al. (2025b) propose an iterative SFT–RL pipeline to reduce instability and cost. RIPT-VLA (Tan et al., 2025) applies RLOO (Ahmadian et al., 2024) for online training, RL4VLA (Liu et al., 2025) studies RL-driven generalization, VLA-RL (Lu et al., 2025a) applies PPO, and RFTF (Shu et al., 2025) introduces value models for dense reward estimation. SimpleVLA-RL (Li et al., 2025) extends veRL to VLA models with GRPO-based online RL, demonstrating significant improvements in data efficiency, long-horizon task performance, and generalization across spatial, object, and task distributions. RLinf (Yu et al., 2025) provides a scalable and unified pipeline for VLA RL—combining rendering, inference, and training—to boost efficiency and performance. $\pi_{\mathrm{RL}}$ (Chen et al., 2025a) applied online RL fine-tuning for flow-based VLAs. $\pi_{0.6}^*$ (Intelligence et al., 2025) uses RECAP to fine-tune VLAs with advantage-conditioned policies, learning from autonomous experience and expert corrections. Despite their promise, these methods typically optimize at the trajectory level, suffering from sparse rewards, coarse credit assignment, and difficult exploration in long-horizon manipulation. In contrast, our stage-aware RL decomposes trajectories into semantically meaningful stages and assigns stage-level rewards, providing denser, interpretable feedback and enabling progressive optimization for complex robotic tasks.

## 3 PRELIMINARY

### 3.1 PROBLEM FORMULATION

We consider a language-conditioned POMDP problem defined by the tuple $\{\mathcal{S}, \mathcal{A}, \mathcal{T}, \mathcal{L}, \mathcal{R}, \gamma\}$, where $\mathcal{S}$ is the state space, $\mathcal{A}$ is the action space, $\mathcal{T} : \mathcal{S} \times \mathcal{A} \to \mathcal{S}$ is the dynamic function, $\mathcal{L}$ is the space of language instruction, $\mathcal{R} : \mathcal{S} \times \mathcal{L} \to \mathbb{R}$ is the reward function, and $\gamma$ is a scale factor with $0 < \gamma < 1$. The goal of a VLA model is to find a policy $\pi_\theta : \mathcal{S} \times \mathcal{L} \to \mathcal{A}$, which generates action trajectories maximizing the expected accumulated reward, or return for each task $l$, i.e. $\mathcal{R}(\pi, l) = \mathbb{E}_{a \sim \pi}[\sum_t \gamma^t r_t]$.

Fine-tuning a VLA model adapts a pre-trained $\pi_\theta$ to new tasks so that the resulting policy $\pi_{\theta'}$ maximizes expected return under the POMDP. This can be done through imitation for aligning with expert demonstrations, preference for refining trajectories via learned comparisons, or reinforcement optimizing long-term rewards.

### 3.2 TRAJECTORY-WISE PREFERENCE OPTIMIZATION (TPO)

Direct Preference Optimization (DPO) (Rafailov et al., 2023) is a recent fine-tuning technique for large language models that directly aligns a policy with preference data, bypassing explicit reward modeling. Extending this idea to fine-tuning VLA models yields TPO (Zhang et al., 2024): the outputs are action trajectories $\tau = \{(s_t, a_t)\}_{t=1}^T$ rather than text sequences. TPO treats each trajectory as a single sequence and learns from pairwise comparisons of successful and failed trajectories $(\tau^+, \tau^-)$ generated under the same instruction. The policy is updated to prefer $\tau^+$ over $\tau^-$ by minimizing

$$L_{\text{TPO}}(\theta) = -\mathbb{E}_{(\tau^+, \tau^-)}\Big[ \log \sigma\big(\beta(q(\tau^+) - q(\tau^-))\big)\Big], \tag{1a}$$

$$q(\tau) = \frac{1}{T}\sum_{t=1}^T \Big( \log \pi_{\theta'}(a_t|s_t) - \log \pi_\theta(a_t|s_t) \Big). \tag{1b}$$

where $\sigma(\cdot)$ is the sigmoid function, $\beta$ controls the strength of preference alignment, $s_t \in \mathcal{S}$ and $a_t \in \mathcal{A}$ denote the environment state and action at timestep $t$, and $q(\cdot)$ measures the normalized log-likelihood ratio of a trajectory under policy $\pi_{\theta'}$ relative to $\pi_\theta$. $L_{\text{TPO}}$ is minimized when the model increases $q(\tau^+)$ relative to $q(\tau^-)$, i.e., when the likelihood of successful trajectories exceeds failed ones.

While TPO provides a direct mechanism to apply preference learning to long-horizon control, it suffers from credit assignment ambiguity: preferences are assigned to full trajectories, making it difficult to determine which specific stage contributed to the preference signal. Moreover, such a binary preference limits optimization to coarse distinctions between successful and failed rollouts, without capturing relative quality among partially successful trajectories. These limitations motivate (STA-TPO), which decomposes trajectories into stages and aligns hierarchical preferences at the stage level, enabling finer-grained optimization.

### 3.3 PROXIMAL POLICY OPTIMIZATION (PPO)

PPO (Schulman et al., 2017) is one of the most widely used online reinforcement learning algorithms, known for its balance of sample efficiency and training stability. PPO improves policy gradient methods by introducing a clipped surrogate objective that prevents excessively large policy updates, thereby stabilizing training. Given an old policy $\pi_\theta$, the clipped objective is

$$L_{\text{PPO}}(\theta) = \mathbb{E}_t\Big[ \min\big(p_t(\theta)\,\text{GAE}(r_t),\ \text{clip}(p_t(\theta), 1 - \epsilon, 1 + \epsilon)\,\text{GAE}(r_t)\big)\Big], \tag{2}$$

where $p_t(\theta) = \pi_{\theta'}(a_t|s_t)/\pi_\theta(a_t|s_t)$ is the likelihood ratio between the new and old policies, and $\epsilon$ is a clipping parameter. $\text{GAE}(\cdot)$ is generalized advantage estimator (Schulman et al., 2015) that estimate the advantage value based on rewards $r_t$.

In the context of fine-tuning VLA models, PPO is commonly used to fine-tune policies with sparse $r_t$, but such signals often limit sample efficiency and provide insufficient guidance for complex, long-horizon tasks. This motivates STA-PPO, which integrates stage-aware reward shaping to transform sparse terminal rewards into dense progressive signals for more efficient fine-tuning.

# 4 METHOD

We begin by introducing *Stage-Aware Reinforcement* (STARE), which decomposes long-horizon action trajectories into semantically meaningful stages, each equipped with stage-wise costs and intra-stage rewards. Building on this foundation, we develop offline and online learning algorithms for progressive stage-wise finetuning: *Stage-Aware Trajectory Preference Optimization* (STA-TPO) and *Stage-Aware Proximal Policy Optimization* (STA-PPO). Finally, we integrate STA-TPO and STA-PPO with supervised fine-tuning (SFT) into a serial pipeline, *Imitation→Preference→Exploration* (IPI), to achieve sufficient fine-tuning of VLA models.

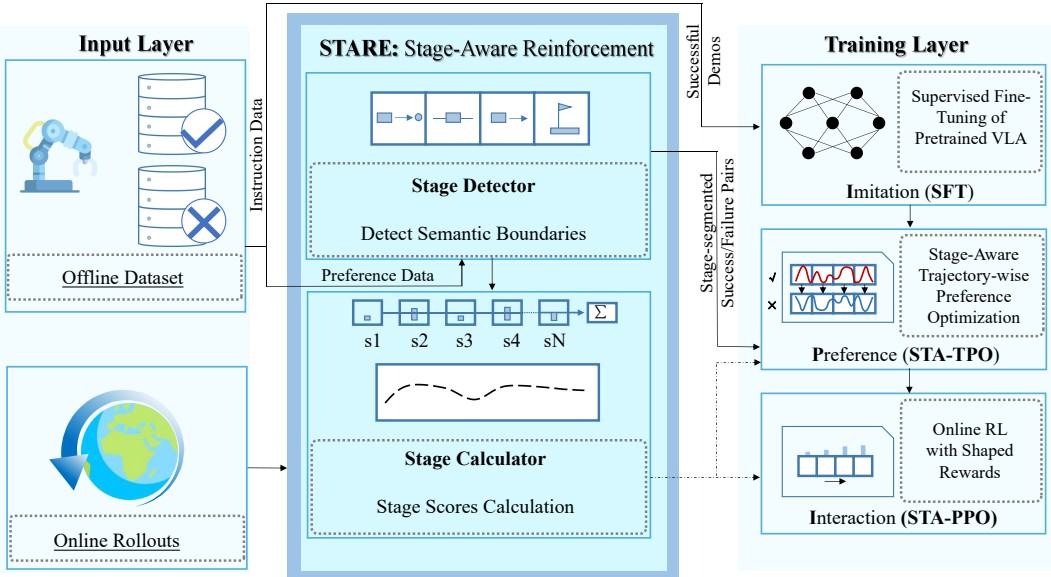

Figure 2: Overview of the STARE Framework and Its Integration into the IPI Training Pipeline.

## 4.1 STAGE-AWARE REINFORCEMENT (STARE)

We propose STARE, a module that decomposes long-horizon action trajectories into semantically meaningful stages defined by task-specific rules. STARE consists of two components: (i) a *stage separator*, which determines *when* stage transitions occur by detecting task-relevant events, and (ii) a *stage calculator*, which evaluates *how well* each stage is executed using stage-wise costs and dense intra-stage rewards.

**Stage Separator** Stage boundaries are determined by semantically meaningful manipulation events rather than arbitrary temporal cuts. Given the whole action trajectory $\tau$, we intend to divide it into $K$ stages by defining semantic boundaries and assigning each global timestep $t$ a stage label $k \in \{1, \ldots, K\}$. Following an event-driven rule, the entry condition of stage $k$ coincides with the terminal condition of stage $k - 1$, ensuring progressive continuity across stages. For instance, a pick-and-place task can be separated into four stages: *Reach → Grasp → Transport → Place*.

Stage segmentation thus reduces to detecting the onset of each stage based on geometric constraints, defined by thresholds $\delta_k$ on the translation and orientation signals of the end-effector. These thresholds set binary environment flags (e.g., grasped, on-target). For example: a *Reach → Grasp* transition occurs when the end-effector contacts the object; *Grasp → Transport* occurs when the grasped object is lifted above a small height threshold; *Transport → Place* occurs when the object is within a distance margin of the goal position; and *Place → Success* occurs when the object is released and remains stably in the goal region (see Supplementary Material E for other segmentation examples). Thereby, we group steps with the same stage label into the $k$-th trajectory segment $\tau^{(k)} = \{(s_t, a_t) \mid g(t) = k\}_{t=1}^{T_k}$, where $s_t \in \mathcal{S}$, $a_t \in \mathcal{A}$, and $T_k$ is the number of timesteps assigned to stage $k$. Here, $g : \mathbb{N} \to \{1, \ldots, K\}$ is a stage assignment function mapping each timestep

$t$ to its corresponding stage index $k$. The full trajectory can then be expressed as the stage-wise decomposition $\tau \mapsto \{\tau^{(i)}\}_{i=1}^{K}$.

**Stage Calculator** Given the stage segments produced by the stage separator, the stage calculator computes both stage-wise costs and intra-stage dense rewards by measuring the relation between the end-effector and relevant targets in the environment. The specific forms of cost and reward depend on the goal of each stage. We illustrate with *Reach* as the $k$-th stage:

(i) *Stage cost aggregation.* We define the cost function $\ell_k(\cdot)$ as the mean Euclidean distance over $T_k$ between the end-effector and the target object from start to the end of *Reach*:

$$\ell_k(\tau^{(k)}) = \frac{1}{T_k} \sum_{t=1}^{T_k} \|x_{\text{ee}}(t) - x_{\text{obj}}(t)\|_2, \tag{3}$$

where $x_{\text{ee}}(t) \in \mathbb{R}^3$ denotes the Cartesian position of the end-effector at time step $t$, and $x_{\text{obj}}(t) \in \mathbb{R}^3$ is the target position of the object. By definition, $\ell_k$ is a non-negative value measuring the deviation from the target: the better the $\tau^{(k)}$, the smaller the $\ell_k$. Detailed cost functions for other stage categories are provided in the Supplementary Material D.

(ii) *Intra-stage reward shaping.* To provide dense guidance, we adopt potential-based reward shaping (Kim et al., 2025; Ng et al., 1999). For active stage $k$, we define a per-timestep potential $\Phi_{k_t}$ that captures the normalized progress of state $s_t$. Specifically, for *Reach*, we use:

$$\Phi_{k_t}(s_t) = \sigma\left(1 - \frac{\|x_{\text{ee}(t)} - x_{\text{obj}(t)}\|}{d_k}\right), \tag{4}$$

where $\Phi_{k_t}(s_t) \in [0, 1]$, $d_k$ is a normalization length scale, and $\sigma(\cdot)$ is a sigmoid function. This provides smooth shaping rewards that encourage the end-effector to progressively reach the target (see detailed potential functions for other stages in the Supplementary Material E). Based on $\Phi_{k_t}$, the shaped reward $r'_t$ augments the sparse reward $r_t$ as:

$$r'_t = r_t + \gamma \Phi_{k_{t+1}}(s_{t+1}) - \Phi_{k_t}(s_t). \tag{5}$$

## 4.2 FROM STARE TO STA-TPO

Unlike standard TPO (Zhang et al., 2024), which aggregates preferences only at the level of entire trajectories, STA-TPO leverages STARE to segment trajectories into progressive stages and perform stage-wise preference alignment. A detailed algorithm is shown in Supplementary Material A.1. A pair comparison of stage samples $(\tau^{(k)+}, \tau^{(k)-})$ exists only when the previous stage $\tau^{(k-1)}$ has been successfully completed, ensuring progressive consistency across stages. In addition, the stage cost $\ell_k(\tau)$ is incorporated as a penalty term in equation 1b, transforming $q$ into $\hat{q}$:

$$\hat{q}(\tau^{(k)}) = q(\tau^{(k)}) - \lambda \ell_k(\tau^{(k)}), \tag{6}$$

where $\lambda$ is the penalty weight. The original objective $\mathcal{L}_{\text{TPO}}$ in equation 1a thereby extends to $\mathcal{L}_{\text{STA-TPO}}$. Compared to equation 1, which optimizes the model only with binary trajectory-level preferences (success vs. failure), $\hat{q}$ introduces a hierarchical signal to $\mathcal{L}_{\text{STA-TPO}}$. Even among successful stages $\tau^{(k)+}$ across different trajectories, those with lower penalties $\ell_k(\tau^{(k)})$ yield higher $\hat{q}$, while less optimal stages receive lower $\hat{q}$. This design enables credit assignment not only between success and failure but also among varying degrees of success, thereby providing finer-grained supervision for learning optimal behaviors.

## 4.3 FROM STARE TO STA-PPO

For online RL fine-tuning, we integrate STARE directly into rollouts. The stage separator determines the stage transition on the fly. At each time step within the stage, the stage calculator produces shaped reward $r'_t$, turning $L_{\text{PPO}}$ in equation 2. to $L_{\text{STA-PPO}}$. Finally, policy parameters $\theta$ are updated by minimizing $L_{\text{STA-PPO}}$. By replacing $r_t$ with $r'_t$, STA-PPO provides denser, stage-aligned feedback that accelerates policy learning in long-horizon, sparse-reward tasks. A detailed algorithm is shown in Supplementary Material A.2.

### 4.4 STA-TPO AND STA-PPO FOR SERIAL FINE-TUNING

Existing works often apply offline preference-based optimization (Zhang et al., 2024) and online RL fine-tuning (Liu et al., 2025; Li et al., 2025) separately. Besides, while we are now able to jointly address offline preference alignment and online reinforcement learning by STA-TPO and STA-PPO, a complete fine-tuning framework for VLA models must also incorporate imitation learning to initialize a strong policy prior.

Thereby, we propose Imitation→Preference→Interaction (IPI), a three-step fine-tuning pipeline. We first warm up the policy safely from demonstrations by SFT. Then we apply STA-TPO to offline refine the policy. Finally, we apply STA-PPO to further enhance robustness through online exploration. Thereby, IPI integrates supervised, preference-based, and exploration signals into a coherent progression, yielding more sample-efficient and more robust fine-tuning of VLA models.

## 5 EXPERIMENTS

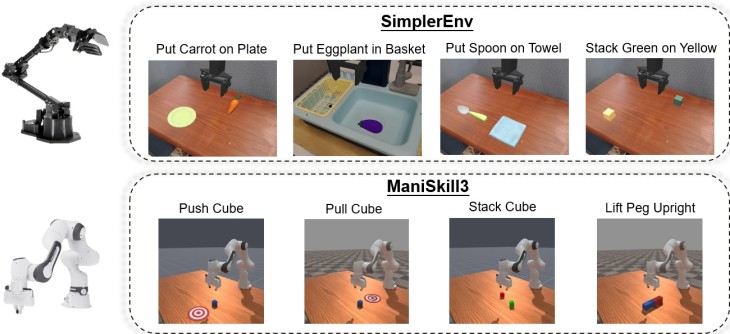

Figure 3: Two simulated benchmarks. We show experiment setups and example tasks involved.

**Benchmarks & Baselines.** We evaluate our approach on two families of robotic manipulation environments, as shown in Figure 3. The first is **SimplerEnv** (Li et al., 2024b) with the WidowX arm, where we focus on the four canonical single-object tasks in the **SimplerEnv-WidowX** split. The second is **ManiSkill3** (Tao et al., 2025) with the Franka robot (Haddadin, 2024), including *StackCube* and three contact-rich tasks (*PushCube*, *PullCube*, and *LiftPegUpright*) to validate generality beyond pick-and-place and assess performance under challenging non-trivial manipulation. We compare against widely used VLA baselines (RT-1-X, Octo-Base/Small, RoboVLM, SpatialVLA), the strong offline preference fine-tuning method *GRAPE*, and the RL fine-tuning baseline *RL4VLA* (Liu et al., 2025). For fairness, all methods are fine-tuned on the same two backbones: OpenVLA-7B (Kim et al., 2024) and pi0.5_base (Black et al., 2025), and we additionally evaluate our proposed STA-TPO, STA-PPO, and the full **IPI**. We report average success over 300 evaluation episodes per method and setting. Unless otherwise stated, hyperparameters are shared across methods when applicable (detailed in Appendix).

**Main Results.** We begin by presenting overall comparisons on two widely used families of manipulation benchmarks. Table 1 reports grasp and final success rates on the four representative **SimplerEnv-WidowX** tasks, while Table 2 reports results on selected **ManiSkill3-Franka** tasks including both stacking and contact-rich manipulation.

Across all benchmarks, existing VLA baselines exhibit limited performance (e.g., average success rates $< 60\%$). Recent RL fine-tuning approaches, such as RL4VLA (Liu et al., 2025) achieve strong results (92.6% on SimplerEnv-WidowX, 70.5% on ManiSkill3). Our proposed **IPI** further improves to 98.0% and 96.4%, outperforming prior state-of-the-art methods by +5.4 and +25.9 points, nearly solving these benchmark tasks. Our **IPI** is a fully implemented and executed pipeline obtained from actual end-to-end runs, demonstrating that each stage can be integrated seamlessly and that the complete framework achieves the strongest overall performance.

Table 1: **Evaluation on SimplerEnv with WidowX Robot tasks.** We report the final success rate and grasp success rate shown in parentheses (Success % (Grasp %)). Our method, IPI, uses RL fine-tuning after an initial SFT phase. The '(+X%)' indicates the improvement over a relevant baseline.

| Methods | Robotic Task | | | | Avg. Success Rate (%) |
|---|---|---|---|---|---|
| | Put Spoon on Towel | Put Carrot on Plate | Stack Green on Yellow | Put Eggplant in Basket | |
| *Other Methods* | | | | | |
| RT-1-X (Brohan et al., 2023) | 0.0 (16.7) | 4.2 (20.8) | 0.0 (8.3) | 0.0 (0.0) | 1.1 |
| Octo-Base (Ghosh et al., 2024) | 12.5 (34.7) | 8.3 (52.8) | 0.0 (31.9) | 43.1 (66.7) | 16.0 |
| Octo-Small (Ghosh et al., 2024) | 47.2 (77.8) | 9.7 (27.8) | 4.2 (40.3) | 56.9 (87.5) | 30.0 |
| RoboVLM (Li et al., 2024a) | 20.8 (37.5) | 25.0 (33.3) | 8.3 (8.3) | 0.0 (0.0) | 13.5 |
| SpatialVLA (Qu et al., 2025) | 20.8 (25.0) | 20.8 (41.7) | 25.0 (58.3) | 70.8 (79.2) | 34.4 |
| SOFAR (Qi et al., 2025) | 58.3 (62.5) | 66.7 (75.0) | 70.8 (91.7) | 37.5 (66.7) | 58.3 |
| *OpenVLA-7B Based Methods* | | | | | |
| SFT | 43.7 (70.3) | 52.7 (74.7) | 21.3 (59.0) | 49.0 (67.3) | 41.7 |
| GRAPE (Zhang et al., 2024) | 44.3 (72.0) | 55.0 (85.3) | 22.7 (53.3) | 53.7 (78.7) | 43.9 |
| SFT → STA-TPO (**STARE**) | 51.0 (85.7) | 57.3 (82.3) | 43.7 (78.3) | 54.3 (85.7) | 51.6 (+17.7) |
| RL4VLA (Liu et al., 2025) | 93.0 (98.3) | 91.3 (96.7) | 92.0 (97.0) | 93.7 (98.3) | 92.5 |
| SFT → STA-PPO (**STARE**) | 94.3 (97.7) | 95.3 (99.0) | 93.7 (98.3) | 95.0 (98.7) | 94.6 (+2.1) |
| **IPI (STARE)** | **98.0)** (99.0) | **98.5** (99.5) | **98.0** (99.0) | **97.5** (99.0) | **98.0** |
| *Pi0.5 Based Methods* | | | | | |
| SFT | 49.3 (85.3) | 64.7 (89.3) | 44.7 (76.0) | 69.7 (92.3) | 57.1 |
| GRAPE (Zhang et al., 2024) | 48.0 (78.7) | 59.3 (88.3) | 48.3 (69.7) | 58.7 (80.3) | 53.6 |
| SFT → STA-TPO (**STARE**) | 54.0 (83.7) | 65.3 (80.3) | 54.0 (70.7) | 68.7 (83.7) | 60.5 (+6.9) |
| $\pi_{RL}$ (Chen et al., 2025a) | 82.7 (98.3) | 97.3 (99.3) | 83.3 (97.3) | 55.0 (69.7) | 79.6 |
| SFT → STA-PPO (**STARE**) | 90.7 (98.7) | 97.7 (99.0) | 85.7 (97.3) | 63.7 (85.7) | 84.5 (+4.9) |
| **IPI (STARE)** | **95.7** (99.3) | **98.7** (99.7) | **93.0** (98.0) | **78.7** (91.0) | **91.5** |

**PPO vs STA-PPO** While PPO improves over SFT, it often stagnates on tasks requiring high precision or contact-rich interactions. In contrast, STA-PPO consistently accelerates convergence and achieves higher asymptotic performance by leveraging stage-aware signals. Figure 4 presents results across eight representative tasks from SimplerEnv-WidowX and ManiSkill3. The most challenging tasks—*LiftPegUpright* and *StackGreenOnYellow*—exhibit the largest performance gaps, underscoring the importance of incorporating stage-aware signals in long-horizon, precision-critical manipulation. By comparison, for short-horizon pick-and-place tasks (e.g., *PutCarrotOnPlate*, *PutEggplantInBasket*) or simple push-and-pull tasks, PPO and STA-PPO achieve similar final success rates, with STA-PPO mainly contributing faster convergence and reduced variance. Overall, these results suggest that stage-aware guidance is particularly crucial when strict alignment accuracy or multi-stage coordination is required, whereas simpler tasks can often be solved effectively with standard reinforcement learning.

We compare SFT, PPO, STA-PPO, and our full IPI method. While PPO improves over SFT, it often stagnates in high-precision (e.g., *StackCube)* or contact-rich settings (e.g., *LiftPegUpright*). STA-PPO accelerates convergence and achieves higher asymptotic performance by leveraging stage-aware signals. Notably, the most challenging tasks, *LiftPegUpright* and *StackCube*, show the clearest benefit, highlighting the importance of incorporating stage-awareness for solving complex tasks.

After the benchmark-level comparison in Tables 1 and 2, we note that while the overall improvements of STA-PPO and STA-TPO over prior baselines are consistent, the performance gap is most pronounced on two tasks: (1) Cube stacking tasks from both the envs, which requires precise alignment in placing stage, and (2) *LiftPegUpright* from ManiSkill3, which demands accurate orientation control after lifting. To better understand where these gains originate, we decompose trajectories into semantic stages and evaluate **conditional stage success** ($P(\text{stage}_k \mid \text{stage}_{k-1})$), which measures how reliably a policy completes a stage given that all previous stages have been successful.

Figure 5 shows STA-TPO provides clear advantages over TPO, with the largest improvements appearing in the **grasp**, and **place** and **upright** stages. These stages are particularly decisive for the

Table 2: **Evaluation on selected ManiSkill3 Franka tasks.** RoboFAC-7B is a VLM method Lu et al. (2025b), Octo and SmolVLA use 1000 trajectory samples for SFT per task while OpenVLA-7B and Pi0.5 based methods use 100 trajectory samples for SFT and 50 trajectory preference pairs for TPO and STA-TPO, detailed in Appendix C.2.

| Methods | Robotic Task | | | | Avg. Success Rate (%) |
|---|---|---|---|---|---|
| | Stack Cube | Push Cube | Pull Cube | LiftPeg Upright | |
| *Other Methods* | | | | | |
| Octo (fine-tuning) (Ghosh et al., 2024) | 1.0 | 67.0 | 90.0 | 0.0 | 39.5 |
| SmolVLA (fine-tuning) (Shukor et al., 2025) | 12.7 | 86.3 | 90.7 | 16.3 | 51.5 |
| RoboFAC-7B (Lu et al., 2025b) | 85.5 | 80.4 | 80.7 | 84.0 | 82.7 |
| *OpenVLA-7B Based Methods* | | | | | |
| SFT | 12.0 | 11.7 | 31.0 | 5.3 | 15.0 |
| GRAPE (Zhang et al., 2024) | 15.7 | 13.3 | 35.3 | 7.7 | 18.0 |
| SFT→STA-TPO (**STARE**) | 19.3 | 16.0 | 35.7 | 12.3 | 20.8 (+2.8) |
| RL4VLA (Liu et al., 2025) | 64.0 | 95.7 | 90.3 | 32.0 | 70.5 |
| SFT→STA-PPO (**STARE**) | 92.7 | 96.0 | 95.3 | 89.7 | 93.4 (+22.9) |
| **IPI (STARE)** | **94.3** | **97.3** | **98.5** | **95.5** | **96.4** |
| *Pi0.5 Based Methods* | | | | | |
| SFT | 26.3 | 18.3 | 43.0 | 10.7 | 25.6 |
| GRAPE (Zhang et al., 2024) | 22.7 | 16.3 | 45.0 | 6.3 | 22.6 |
| SFT→STA-TPO (**STARE**) | 28.0 | 22.3 | 44.7 | 15.7 | 27.7 (+5.1) |
| $\pi_{RL}$ (Chen et al., 2025a) | 72.3 | 96.7 | 93.3 | 58.0 | 80.1 |
| SFT→STA-PPO (**STARE**) | 80.7 | 98.0 | 92.7 | 75.3 | 86.7 (+6.6) |
| **IPI (STARE)** | **84.3** | **99.3** | **95.0** | **80.7** | **89.9** |

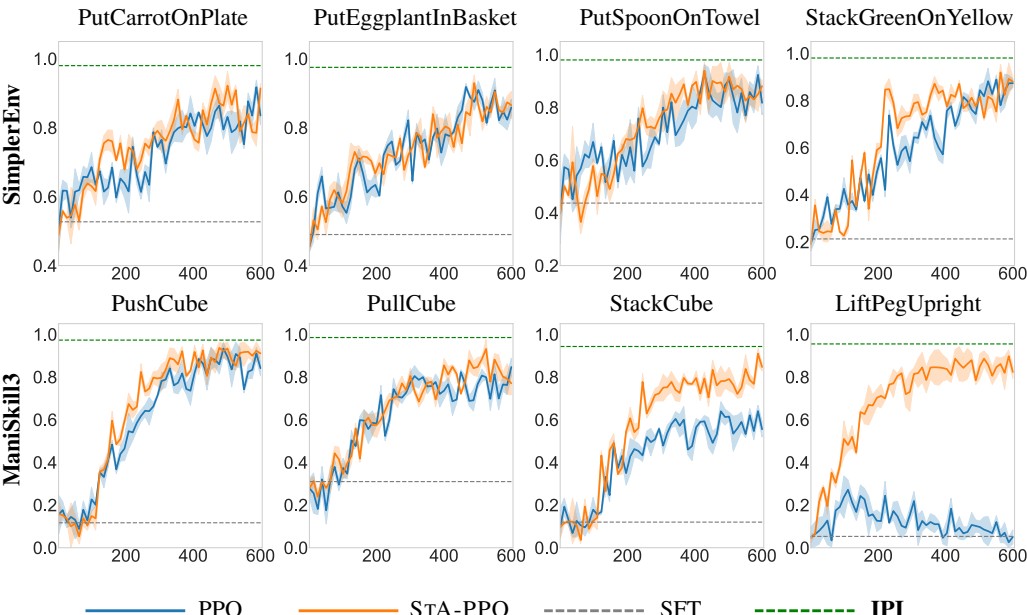

Figure 4: Comparison of learning curves across eight representative tasks from SimplerEnv and ManiSkill3. The y-axis denotes the success rate, and the x-axis shows the interaction environment steps (in thousands).

final outcome, explaining why the overall success rate improvements are disproportionately large for these two tasks.

**Ablations study** To further dissect the contributions of stage-aware reinforcement, we conduct a stage toggle ablation where the STA signal is selectively removed at different phases of the manipulation in STA-PPO. As shown in Figure 6, disabling STA at early stages (e.g., reach or grasp) only leads to moderate drops, since later corrective actions can partially recover performance. In con-

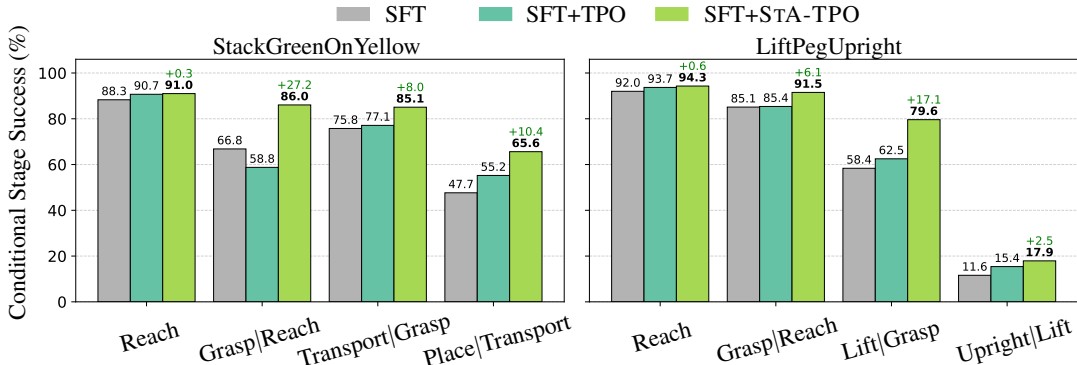

Figure 5: **Offline Stage-wise ablation on two tasks.** We report stage completion rates (%) for *StackGreenonYellow* (SimplerEnv) and *LiftPegUpright* (ManiSkill3). Compared with TPO, STA-TPO achieves significant gains, particularly in the **grasp** and **place/upright** stages, which are critical for final success.

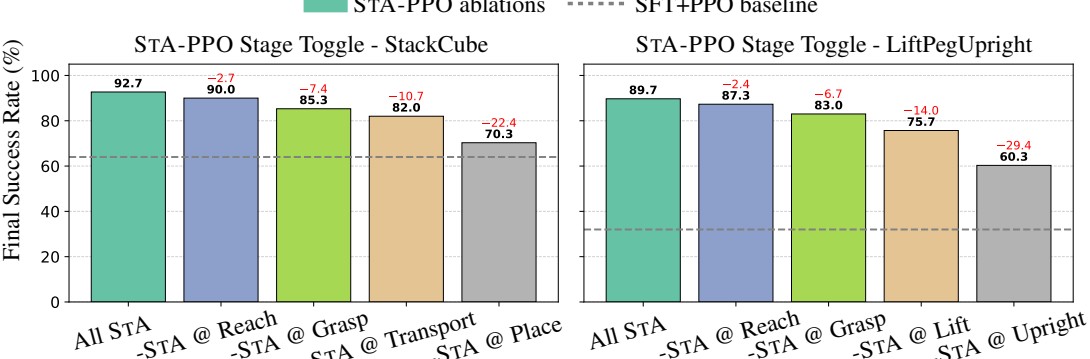

Figure 6: **Stage toggle ablation of STA-PPO.** We evaluate the effect of selectively disabling stage-aware reinforcement signals on two representative tasks: *StackCube* (ManiSkill3) and *LiftPegUpright* (ManiSkill3). The **All STA** setting achieves the best performance, while disabling critical stages (**Place** in stacking, **Upright** in peg lifting) causes the largest performance drops.

trast, removing STARE at the final precision-critical phases (e.g., **Place** in stacking and **Upright** in peg lifting) causes the largest degradation, reducing success rates by more than 20%. This analysis highlights that STA guidance is especially valuable at stages where geometric accuracy and stability directly determine task completion.

## 6 CONCLUSION

We presented *Stage-Aware Reinforcement* (STARE), a plug-in module that decomposes trajectories into semantically meaningful stages and provides stage-level reinforcement signals. Building on this, we introduced Stage-Aware TPO (STA-TPO) and PPO (STA-PPO) for offline stage-wise preference alignment and online intra-stage interaction, and integrated them with supervised fine-tuning into the *Imitation→Preference→interaction* (IPI) pipeline. Experiments on SimplerEnv and ManiSkill3 demonstrate that IPI achieves state-of-the-art success rates and improved generalization. Our results highlight the importance of stage-aware credit assignment for efficient VLA fine-tuning and point toward promising directions in long-horizon robotic learning.

**Reproducibility Statement** We have made extensive efforts to ensure the reproducibility of our work. We detail our proposed module, STARE, in Sec. 4.1. Then we illustrate how we apply STARE in Sec. 4.2 and Sec. 4.3 to yield STA-TPO and STA-PPO. We provided cost functions and potential functions used in STA-TPO and STA-PPO in the Supplementary Material D and E. We also provide algorithms of STA-TPO and STA-PPO in the Supplementary Material A. Training settings and evaluation protocols are provided in the Supplementary Material B. All datasets used (SimplerEnv and ManiSkill3) are publicly available. We provide experimental visualizations and more at https://sites.google.com/view/stare-vla. Finally, we will provide our code and instructions upon acceptance.

**LLM Usage** We used ChatGPT solely as a writing assistant for grammar checking and language polishing of the manuscript. All research ideas, experimental design, implementations, analyses, and conclusions were conceived and conducted entirely by the authors. Separately, our method employs OpenVLA (Kim et al., 2024) (which includes LLM backbones (Karamcheti et al., 2024)) as a standard pretrained model component for VLA modeling. This use is analogous to employing any pretrained backbone and does not involve LLM assistance in research ideation or manuscript drafting. No LLMs were used to generate technical content or to determine experimental methodology.

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

# Supplementary Material

We provide experimental visualizations and more at https://sites.google.com/view/stare-vla.

## A    DETAILED ALGORITHMS

---

**Algorithm 1** STA-TPO (offline)

---

**Require:** Preference pairs $\{(\tau^+, \tau^-)\}$ under same instruction; reference policy $\pi_\theta$; STARE; stage penalty weight $\lambda$; temperature $\beta$; learning rate $\eta$.
**Ensure:** Updated policy $\pi_{\theta'}$.
1: **while** not converged **do**
2:     Sample minibatch $\mathcal{B} = \{(\tau^+, \tau^-)\}$.
3:     **for all** $\tau \in \{\tau^+, \tau^-\}$ in $\mathcal{B}$ **do**
4:         **Stage segmentation & costs:** $\{\tau^{(k)}, \ell_k(\tau^{(k)})\}_{k=1}^K \leftarrow$ STARE$(\tau)$.
5:         **Stage scores:** For each $k$, first compute

$$q(\tau^{(k)}) \leftarrow \frac{1}{T_k} \sum_{t \in T_k} \Big( \log \pi_{\theta'}(a_t|s_t) - \log \pi_\theta(a_t|s_t) \Big). \qquad \text{(cf. equation 1b)}$$

6:         Then add stage costs: $\hat{q}(\tau^{(k)}) \leftarrow q(\tau^{(k)}) - \lambda\, \ell_k(\tau^{(k)})$.
7:     **end for**
8:     **Preference loss:** Compute

$$L_{\text{STA-TPO}} \leftarrow -\frac{1}{|\mathcal{B}|} \sum_{(\tau^+, \tau^-) \in \mathcal{B}} \frac{1}{K} \sum_{k=1}^K \log \sigma\Big(\beta\big(\hat{q}(\tau^{(k)+}) - \hat{q}(\tau^{(k)-})\big)\Big). \qquad \text{(cf. equation 1a)}$$

9:     **Policy update:** $\theta' \leftarrow \theta' - \eta\, \nabla_{\theta'} L_{\text{STA-TPO}}$.
10: **end while**

---

**Algorithm 2** STA-PPO (online)

---

**Require:** Simulation Env; Behavior policy $\pi_\theta$; STARE; horizon $T$; PPO epochs $E$; discount $\gamma$; GAE parameter; clip $\epsilon$; step size $\eta$.
**Ensure:** Updated policy $\pi_{\theta'}$.
1: **while** not converged **do**
2:     **Rollout:** collect $\{(s_t, a_t, r_t, \log \pi_\theta(a_t|s_t))\}_{t=0}^{T-1}$ in Env.
3:     **Online stage labels & potentials:**
4:     **for** $t = 0$ to $T - 1$ **do**
5:         Detect current stage $k = g(t)$ via stage separator in STARE (event rules).
6:         Compute potential $\Phi_{k_t}(s_t)$ by the stage calculator in STARE.
7:     **end for**
8:     **Shaped rewards:**
9:     **for** $t = 0$ to $T - 1$ **do**
10:         $r_t' \leftarrow r_t + \gamma\, \Phi_{k_{t+1}}(s_{t+1}) - \Phi_{k_t}(s_t)$                    ▷ Potential-based shaping
11:     **end for**
12:     **for** $e = 1$ to $E$ **do**                                         ▷ PPO updates
13:         Compute ratio $p_t(\theta') = \exp(\log \pi_{\theta'}(a_t|s_t) - \log \pi_\theta(a_t|s_t))$.
14:         **Interaction loss:**

$$L_{\text{STA-PPO}} \leftarrow \mathbb{E}_t\big[\min\big(p_t(\theta')\, \text{GAE}(r_t'),\; \text{clip}(p_t(\theta'), 1 - \epsilon, 1 + \epsilon)\, \text{GAE}(r_t')\big)\big].$$
$$\text{(cf. equation 2)}$$

15:         **Update:** $\theta' \leftarrow \theta' + \eta\, \nabla_{\theta'} L_{\text{STA-PPO}}$.
16:     **end for**
17: **end while**

---

## B  Training Settings and Evaluation Protocols

### B.1  Supervised Fine-Tuning (SFT)

We initialize from the pretrained OpenVLA backbone and optimize the action prediction objective with AdamW. The learning rate is set to 1e-5 and training runs for several tens of thousands of steps.

### B.2  Trajectory Preference Optimization (TPO / STA-TPO)

For preference optimization, we construct pairs of successful and failed trajectories and train the policy against a frozen reference model. We use AdamW with a learning rate of 2e-5, and train for several epochs. STA-TPO further incorporates stage-wise margins derived from stage calculators.

### B.3  Reinforcement Learning: PPO

We adopt standard PPO with parallel simulation environments. We use discount factor 0.99 and GAE parameter 0.95. Optimization uses AdamW with a policy learning rate of 1e-4 and a value head learning rate of 3e-3. Training proceeds for millions of environment steps.

### B.4  Reinforcement Learning: STA-PPO

STA-PPO shares the same setup as PPO but is augmented with stage-aware signals from the environment. This provides additional supervision during training and stage-level logging during evaluation.

### B.5  Evaluation Protocols

We evaluate on both SimplerEnv-WidowX and ManiSkill3 benchmarks. Each method is tested over 300 evaluation episodes with deterministic action decoding. A trial is considered successful if the environment-defined condition is met within the horizon (60 steps for SimplerEnv and 30 steps for ManiSkill3). We report average success rates over 5 random seeds, and additionally compute conditional stage success to analyze where policies fail or succeed.

## C  Experimental Setup and Implementation Details

### C.1  Environment and Tasks

**SimplerEnv** (Li et al., 2024b) provides a set of real-to-sim environments that enable efficient, scalable, and informative evaluation of robotic policies in simulation, serving as a practical alternative to real-world experiments.

In the **WidowX + Bridge** setting, we evaluate the following tasks:

- **put the spoon on the towel**: The spoon is initially placed at one corner of a 15 cm square region on the tabletop, with a towel located at another corner. The spoon orientation alternates between horizontal and vertical across trials, requiring the robot to adapt its gripper pose. In total, 24 trials are performed.

- **put carrot on plate**: This task mirrors the previous setup, with the spoon replaced by a carrot and the towel replaced by a plate.

- **stack the green block on the yellow block**: A green block and a yellow block (each 3 cm in size) are placed at different corners of a tabletop square. Two square sizes (10 cm and 20 cm side length) are used, resulting in 24 trials.

- **put eggplant into yellow basket**: An eggplant is placed randomly in the right basin of a sink, while a yellow basket is positioned in the left basin. The eggplant's location and orientation vary across trials but remain graspable (avoiding sink edges). A total of 24 trials are conducted.

**ManiSkill3 Tasks** (Tao et al., 2025) are designed as contact-rich robotic manipulation benchmarks with diverse object interactions and long-horizon challenges. In our evaluation, we select the following four tasks with the **Franka** robot:

- **stack cube**: The robot is required to stack one cube on top of another. The cubes are initially placed at separate locations on the table, requiring precise grasping, lifting, and alignment to achieve a stable stack.
- **push cube**: A cube is placed on the tabletop, and the robot must push it toward a designated target region. This task emphasizes contact-rich control and requires smooth trajectory execution.
- **pull cube**: Similar to the push task, but the robot must pull the cube by grasping it from one side and dragging it into the target region, demanding stable grasping under sliding contact.
- **lift peg upright**: The robot is given a peg lying flat on the table. It must first grasp the peg, lift it, and reorient it to stand vertically upright. This task is particularly challenging due to the orientation constraints and precision required to maintain the peg's balance.

## C.2 DATA COLLECTION

Our data collection process, which is similar to the methodology used in the main paper, is designed to generate high-quality data for both supervised and preference-based learning. All data is collected specifically for the selected ManiSkill3 tasks.

- **Expert Demonstrations:** For each task, we generate **100 high-quality demonstration trajectories**. These trajectories are produced using the MPLib motion planner, ensuring kinematically feasible and efficient paths to task completion. Following the findings of the main paper, we apply an action filtering technique to this data, removing idle actions where the end-effector pose changes by a negligible amount. This preprocessing step is crucial for mitigating the issue of trained SFT policies getting stuck during execution.
- **Preference Pairs:** For methods requiring preference data (e.g., TPO), we generate **50 trajectory preference pairs** per task. In the case of SimplerEnv, trajectories are sampled for each task using the Octo model. These pairs are obtained by sampling two trajectories from the Octo-collected dataset, and assigning preference labels based on cumulative rewards (e.g., successful completion vs. failure).

# D STAGE SCORE CALCULATION ACROSS TASKS

**Notation.** A trajectory $\tau = \{(s_t, a_t)\}_{t=1}^{T}$ is segmented into stages $\tau^{(k)}$ with length $T_k = |\tau^{(k)}|$. At time $t$, we denote: - $x_{\text{obj}}(t) \in \mathbb{R}^3$: object position, - $R_{\text{obj}}(t) \in \text{SO}(3)$: object orientation, - $x_{\text{ee}}(t) \in \mathbb{R}^3$: end-effector position, - $x_{\text{goal}}, R_{\text{goal}}$: task goal position/orientation.

We use $\|\cdot\|$ for Euclidean distance and $d_{\text{rot}}(\cdot, \cdot)$ for rotation error.

**Reach.** The average Euclidean distance between the end-effector and the goal:

$$\ell_{\text{Reach}}(\tau) = \frac{1}{T_k} \sum_{t \in \tau^{(k)}} \|x_{\text{ee}}(t) - x_{\text{goal}}\|.$$

**Grasp.** Penalizes poor contact quality and object slipping during grasp. With Contact alignment error between gripper and object $d_{\text{contact}}(t)$ and binary slip flag $\text{slip}(t) \in \{0, 1\}$ (0 = stable, 1 = slipping):

$$\ell_{\text{Grasp}}(\tau) = \frac{1}{T_k} \sum_{t \in \tau^{(k)}} \left( \alpha_1 d_{\text{contact}}(t) + \alpha_2 \, \text{slip}(t) \right).$$

**Transport.** Move the object along a reference trajectory. With reference path $x_{\text{ref}}(t)$:

$$\ell_{\text{Transport}}(\tau) = \frac{1}{T_k} \sum_{t \in \tau^{(k)}} \|x_{\text{obj}}(t) - x_{\text{ref}}(t)\|.$$

**Contact.** (*Push/Pull*) Maintain continuous contact and minimize sideways deviation. With $\text{contact}(t)$ indicating whether the robot maintains contact with the object (1 = contact, 0 = no contact), and $d_\perp(t)$ indicating error orthogonal to the desired push/pull direction:

$$\ell_{\text{Contact}}(\tau) = \frac{1}{T_k} \sum_{t \in \tau^{(k)}} \Big( \alpha_c (1 - \text{contact}(t)) + \alpha_\perp d_\perp(t) \Big).$$

**Push / Pull.** Ensure the object is pushed or pulled straight to the goal. In order to penalize lateral motion that deviates from the desired pushing/pulling direction $\hat{v}$, we define perpendicular motion $\Delta x_\perp(t) = x_{\text{obj}}(t) - x_{\text{obj}}(t-1) - ((x_{\text{obj}}(t) - x_{\text{obj}}(t-1))^\top \hat{v})\hat{v}$:

$$\ell_{\text{Push/Pull}}(\tau) = \|x_{\text{obj}}(T_k) - x_{\text{goal}}\| + \lambda_\perp \frac{1}{T_k} \sum_{t \in \tau^{(k)}} \|\Delta x_\perp(t)\|.$$

**Lift.** Lift the object above a specified minimum height. With current object height $h(t)$ and minimum target height threshold $h_{\min}$:

$$\ell_{\text{Lift}}(\tau) = \frac{1}{T_k} \sum_{t \in \tau^{(k)}} \max(0, h_{\min} - h(t)).$$

**Upright.** Keep the object upright during manipulation. With rotational distance metric $d_{\text{rot}}$ (e.g., geodesic distance on SO(3)) and desired upright orientation $R_{\text{upright}}$:

$$\ell_{\text{Upright}}(\tau) = \frac{1}{T_k} \sum_{t \in \tau^{(k)}} d_{\text{rot}}(R_{\text{obj}}(t), R_{\text{upright}}).$$

**Goal.** Reach the target location while maintaining stability.

$$\ell_{\text{Goal}}(\tau) = \frac{1}{T_k} \sum_{t \in \tau^{(k)}} \big( \beta_x \|x_{\text{obj}}(t) - x_{\text{goal}}\| + \beta_v \|x_{\text{obj}}(t) - x_{\text{obj}}(t-1)\| \big).$$

**Place.** The object must be placed at the goal pose and remain stable for the last $M$ steps:

$$\ell_{\text{Place}}(\tau) = \lambda_{\text{stab}} \frac{1}{M} \sum_{t=T_k-M+1}^{T_k} \|x_{\text{obj}}(t) - x_{\text{obj}}(t-1)\|.$$

**Usage.** Stage costs $\ell_k(\tau)$ define the margin $m_k = \lambda(\ell_k(\tau^-) - \ell_k(\tau^+))$ for SA-TPO, and can also be normalized into potentials for stage-wise reward shaping.

# E  STAGE POTENTIALS FOR DETECTION AND SHAPING

We design stage potentials $\Phi_{\text{stage}}(s)$ to provide both a detection signal (for identifying stage completion) and a shaping signal (for smoother learning). All potentials are bounded in $[0, 1]$ and include tolerance parameters to avoid brittleness to small deviations.

## E.1  PICK–PLACE

This task involves moving an object from its initial location to a designated goal region. We decompose it into four stages: (i) *reach* — the end-effector approaches the object, (ii) *grasp* — establish and maintain a stable grasp, (iii) *transport* — carry the grasped object toward the goal, and (iv) *place* — align and release the object at the goal pose.

$$\Phi_{\text{reach}}(s) = \sigma\Big(1 - \frac{\|p_{\text{ee}} - p_{\text{obj}}\|}{d_{\text{reach}}}\Big),$$

$$\Phi_{\text{grasp}}(s) = \beta_1 \mathbb{1}\{\text{grasped}\} + \beta_2 \tanh\Big(\frac{\text{consec\_grasp}}{\tau}\Big),$$

$$\Phi_{\text{transport}}(s) = \sigma\Big(1 - \frac{\|p_{\text{obj}} - p_{\text{goal}}\|}{d_{\text{trans}}}\Big) + \beta_3 \mathbb{1}\{\text{grasped}\},$$

$$\Phi_{\text{place}}(s) = \sigma\Big(1 - \frac{\|p_{\text{obj}} - p_{\text{goal}}\|}{d_{\text{place}}}\Big) + \beta_4 \, g(R_{\text{obj}}, R_{\text{goal}}),$$

where $p_{\text{ee}}$, $p_{\text{obj}}$, and $p_{\text{goal}}$ are the positions of the end-effector, object, and goal, respectively, and $g(\cdot) \in [0, 1]$ measures orientation alignment between object and goal.

### E.2 PUSH/PULL–CUBE

This task requires pushing a cube along the table surface into a marked goal region. The stages are: (i) *reach* — approach the object, (ii) *contact* — establish end-effector contact, (iii) *push* — move the object toward the goal, and (iv) *goal* — verify the object is inside the goal region.

$$\Phi_{\text{reach}}(s) = \sigma\left(1 - \frac{\|p_{\text{ee}} - p_{\text{obj}}\|}{d_{\text{reach}}}\right),$$

$$\Phi_{\text{contact}}(s) = \beta_1 \mathbb{1}\{\text{ee–obj contact}\},$$

$$\Phi_{\text{push/pull}}(s) = \sigma\left(1 - \frac{\|p_{\text{obj}} - p_{\text{goal}}\|}{d_{\text{push/pull}}}\right),$$

$$\Phi_{\text{goal}}(s) = \mathbb{1}\{\text{obj in goal region}\}.$$

### E.3 LIFT–PEG–UPRIGHT

This task requires picking up a peg, lifting it above the table, and orienting it upright. We decompose it into: (i) *reach* — approach the peg, (ii) *grasp* — secure the peg, (iii) *lift* — raise it to a sufficient height, and (iv) *upright* — align it with the vertical axis.

$$\Phi_{\text{reach}}(s) = \sigma\left(1 - \frac{\|p_{\text{ee}} - p_{\text{peg}}\|}{d_{\text{reach}}}\right),$$

$$\Phi_{\text{grasp}}(s) = \beta_1 \mathbb{1}\{\text{peg grasped}\},$$

$$\Phi_{\text{lift}}(s) = \sigma\left(\frac{z_{\text{peg}} - z_{\text{table}}}{h_{\text{lift}}}\right),$$

$$\Phi_{\text{upright}}(s) = g(R_{\text{peg}}, R_{\text{upright}}),$$

where $z_{\text{peg}}$ is the peg height, $h_{\text{lift}}$ is a normalizing scale, and $g(\cdot) \in [0, 1]$ measures uprightness via cosine similarity between peg orientation and the vertical axis.

**Remark.** All potentials are non-negative by design, with values in $[0, 1]$. Tolerance parameters such as $d_*$ or $h_{\text{lift}}$ define acceptable ranges, ensuring robustness to small deviations and avoiding brittle binary signals.

## F BROADER IMPACTS

Our framework aims to improve safety and reliability of robot manipulation by aligning policies with human-preferred, semantically correct behaviors. Potential risks include overfitting to biased preferences and misuse in unsafe settings; we discuss mitigations in the main text and Appendix.

