# OpenReview forum: "Progressive Stage-Aware Reinforcement for Fine-Tuning Vision-Language-Action Models"
_ICLR.cc/2026/Conference — ICLR 2026 Conference Desk Rejected Submission_

### Official Review · Reviewer_uyNk · 2025-10-28

**Soundness:** 2
**Presentation:** 3
**Contribution:** 2
**Rating:** 4
**Confidence:** 3

**Summary:**

This paper addresses the challenge of directly applying large language model fine-tuning paradigms to Vision-Language-Action (VLA) models, where action trajectories are long-horizon and causally staged, making credit assignment difficult. The authors propose Stage-Aware Reinforcement (STARE), a module that decomposes robot trajectories into semantically meaningful stages and provides dense, stage-level rewards using a stage separator and stage cost calculator. Building on STARE, they introduce Stage-Aware TPO for offline preference alignment and Stage-Aware PPO for online reinforcement learning. These components are unified with supervised imitation learning into a three-phase pipeline called IPI. Experiments on SimplerEnv and ManiSkill3 show state-of-the-art performance with significant improvements in sample efficiency and generalization. Overall, the work demonstrates that stage-aware credit assignment is crucial for efficient fine-tuning of VLA models in long-horizon tasks.

**Strengths:**

- The paper clearly articulates a fundamental mismatch between LLM fine-tuning paradigms and robotic action learning, emphasizing that long-horizon manipulation tasks are composed of semantically distinct, causally ordered stages with varying difficulty. An important observation is well-motivated by Figure 1.

- The proposed IPI pipeline presents a compelling and practical design that unifies supervised imitation learning, offline stage-aware preference alignment, and online reinforcement learning. This consolidation removes the disconnect between earlier approaches and provides a more structured path for VLA fine-tuning.

- By leveraging stage-aware decomposition, the method delivers more precise credit assignment than monolithic trajectory-level optimization, improving sample efficiency and effectively addressing a major bottleneck in long-horizon RL for robotics.

**Weaknesses:**

- While framed as "no additional human labels", the rule design implicitly encodes expert knowledge about the task's subgoals. This may constitute a form of environment-specific engineering, reducing the method's generality compared to approaches that infer task progress from data alone.

- STARE depends on hand-crafted geometric constraints on end-effector and object states to detect stage transitions. This design assumes access to full object states, which may not be available or reliable in many real-world or partially observable scenarios.

- The pipeline heavily relies on correct stage boundaries, yet the paper does not study how segmentation inaccuracies propagate through preference optimization and RL. A sensitivity or ablation analysis (e.g., introducing noise or perturbations to stage boundaries) would strengthen the claims about reliability.

**Questions:**

- How sensitive is STARE to incorrect or noisy stage boundaries?

- Stage detection assumes access to accurate end-effector and object states. How would your approach adapt to partial observability or environments where object status is harder to detect (e.g., the real-world setup)?

- Although STARE avoids explicit human labels, the rules and their thresholds encode expert priors. Could you elaborate on the amount of domain engineering required per new task and whether a generic rule set is feasible?

---

### Official Review · Reviewer_Y7GD · 2025-10-29

**Soundness:** 2
**Presentation:** 3
**Contribution:** 3
**Rating:** 4
**Confidence:** 4

**Summary:**

This paper proposes STARE, a framework that decomposes trajectories into semantically meaningful segments. Each segment is associated with a goal and equipped with a dedicated cost function, effectively transforming sparse rewards into dense ones. Building on this dense reward formulation, the authors integrate STARE with existing RL techniques—TPO and PPO—and report improvements in both offline and online RL settings. Furthermore, the paper enhances the RL pipeline of VLA and introduces IPI, a unified framework that combines offline and online RL within a single stage-aware process.

**Strengths:**

1.	The proposed idea of decomposing trajectories into semantically meaningful segments and designing distinct cost functions for each subtask is well-motivated. This approach provides a principled way to transform sparse rewards into dense rewards, which can substantially improve learning efficiency and policy performance.
2.	The introduction of the new RL training framework, IPI, is novel. By combining offline RL and online RL within a unified framework, the method goes beyond existing pipelines and achieves further performance gains.

**Weaknesses:**

1. Unreasonable experimental design.

a. Task and Benchmark selection. The authors conduct experiments on 8 tasks from Simpler and ManiSkill3. Overall, the difference between STA-PPO and PPO (as reported in RL4VLA) is actually not very large. In the experiments, the authors claim that STA-TPO and STA-PPO show significant advantages in long-horizon tasks (e.g., LiftPegUpright and StackGreenOnYellow, see lines 370–371). In fact, these two tasks are not genuinely long-horizon. For example, StackGreenOnYellow is nearly identical to StackCube in ManiSkill3, yet the authors classify one as long-horizon and the other as short-horizon, which is self-contradictory. A proper definition of long-horizon should at least reference established benchmarks such as LIBERO[1] or VLABench[2], which contain multi-pick-and-place tasks with substantially larger step counts and task diversity. I believe that to highlight the advantage of STARE’s dense reward, it must be demonstrated on truly long-horizon tasks.

b. Baseline Method Selection. I believe the authors need to compare against more RL models. The chosen baselines, GRAPE and RL4VLA, do not involve dense reward design, which means the comparisons are essentially limited to dense-reward TPO and PPO. To demonstrate the soundness of STARE’s reward design, I think it is necessary to at least include comparisons with works such as VLA-RL[3] or TGRPO[4] that explicitly incorporate dense reward, even if the evaluation is limited to a single benchmark like LIBERO.

c. Missing ablation on generality. I would like to see an ablation study where the proposed IPI framework is applied to fine-tune different VLA architectures with varying capacities and performance levels, such as pi0/pifast. Such results would strengthen the claim of STARE’s generality and robustness.

2. Methodological Concerns

a. Rule-based reward dependency. The proposed STARE framework relies on rule-based reward functions designed with privileged information available in simulation. It is unclear how such reward functions would be designed for more complex manipulation tasks, e.g., highly dynamic tasks in RLBench or RL100[5]. The hand-crafted reward design may become infeasible or brittle in such cases.

b. Real-world applicability. While the idea of process reward is a meaningful contribution, robotics ultimately aims at solving real-world problems. The current work is entirely simulation-based. I believe the authors should at least demonstrate one set of offline RL experiments in real-world tasks to verify that the proposed reward design also applies outside simulation. Moreover, the reward functions in this work require significant privileged information (e.g., exact object positions). It is unclear how this sim-to-real gap can be addressed, and how the approach could be made practical without such privileged states.

References:
[1] LIBERO: Benchmarking Knowledge Transfer for Lifelong Robot Learning \
[2] VLABench: A Large-Scale Benchmark for Language-Conditioned Robotics Manipulation with Long-Horizon Reasoning Tasks \
[3] VLA-RL: Towards Masterful and General Robotic Manipulation with Scalable Reinforcement Learning \
[4] TGRPO: Fine-tuning Vision-Language-Action Model via Trajectory-wise Group Relative Policy Optimization \
[5] RL-100: Performant Robotic Manipulation with Real-World Reinforcement Learning

**Questions:**

1. Training process clarity. The authors do not explicitly describe the RL training process. It is unclear whether rollouts are conducted jointly across multiple tasks in a multi-task learning setting, trained sequentially across tasks, or limited to single-task training with single-task evaluation. These setups are fundamentally different and should be clarified.

2. Unclear training details. In the appendix, the number of training steps is not clearly reported. Instead, vague expressions such as “several epochs” are used. This lack of precision in the training setup makes reproducibility difficult for the community.

3. I am curious about how failure trajectories in ManiSkill are collected during preference alignment training.

4. Could the authors provide a success rate or quality analysis of using STARE as a segment separator, explicitly demonstrating the rationality or success rate of each semantically defined process?

5. I would like to know the computational resources and training time consumed by these experiments.

---

### Official Review · Reviewer_D2oS · 2025-10-30

**Soundness:** 2
**Presentation:** 2
**Contribution:** 2
**Rating:** 4
**Confidence:** 3

**Summary:**

This article addresses the issues of coarse-grained credit assignment and unstable training in Vision-Language-Action (VLA) models for long-horizon robotic manipulation. It proposes the Stage-Aware Reinforcement (STARE) module, which decomposes long-horizon action trajectories into semantically meaningful causal stages and generates dense, interpretable stage-aligned reinforcement signals. Building on STARE, the authors further improve offline Trajectory-wise Preference Optimization (TPO) and online Proximal Policy Optimization (PPO) to develop Stage-Aware TPO (STA-TPO) and Stage-Aware PPO (STA-PPO), respectively. Additionally, the article integrates Supervised Fine-Tuning (SFT), STA-TPO, and STA-PPO into a three-step serial fine-tuning pipeline called Imitation→Preference→Interaction (IPI). Experiments conducted on the SimplerEnv (with WidowX arm) and ManiSkill3 (with Franka robot) benchmarks show that IPI achieves state-of-the-art success rates of 98.0% on SimplerEnv and 96.4% on ManiSkill3, significantly outperforming existing baseline methods such as RT-1-X, Octo series, and RL4VLA

**Strengths:**

1. The study accurately identifies the core flaw of existing trajectory-level optimization methods for VLA models—ignoring the causal stage characteristics of action trajectories and the varying difficulty across stages. The proposed STARE module achieves a shift from "coarse trajectory-level supervision" to "fine-grained stage-level supervision" through stage segmentation and fine-grained signal generation, providing a more precise basis for credit assignment, especially for stages requiring high precision such as Grasp and Place

2. The authors select two representative robotic manipulation environments (SimplerEnv and ManiSkill3) covering scenarios such as single-object manipulation and contact-rich tasks. A wide range of baselines are included for comparison, encompassing both traditional VLA models (e.g., RT-1-X, RoboVLM) and mainstream fine-tuning methods (e.g., GRAPE, RL4VLA), with all methods fine-tuned on the OpenVLA-7B backbone to ensure fairness. Moreover, ablation experiments verify the necessity of STARE signals in critical stages (e.g., Place, Upright), enhancing the credibility of the experimental conclusions

3. The STARE module is designed as a plug-and-play component that can be flexibly integrated into existing RL frameworks like TPO and PPO without extensive modifications to the original algorithms. The IPI pipeline follows a clear logical flow—initialization via SFT, offline optimization via STA-TPO, and online refinement via STA-PPO—which aligns with the engineering logic of VLA model fine-tuning and facilitates subsequent deployment in real robotic systems

**Weaknesses:**

1.  In the current field of VLA model fine-tuning, the combined framework of "SFT + offline RL + online RL" has been adopted in multiple studies (e.g., the iterative SFT-RL pipeline proposed by Guo et al., 2025b and the RL-driven generalization method of RL4VLA mentioned in the article). Although the core innovation of this paper—the STARE module—focuses on "stage awareness," the idea of "decomposing trajectories into subtasks/stages" also has precedents in the field of long-horizon robotic RL (e.g., stage-wise dense rewards in DEMO and LLM-generated subgoals in RoboHorizon). The overall method framework lacks strong breakthroughs, leading to a sense of homogenization with existing works

2.  For the stage separator in the STARE module, the specific design logic of the "translation/orientation thresholds of the end-effector" is not elaborated in detail (e.g., how thresholds are adaptively adjusted across different tasks and whether they rely on human experience). Additionally, the selection of potential function parameters (e.g., normalization length scale \(d_k\)) in the stage calculator lacks quantitative analysis, and the impact of parameter sensitivity on experimental results is not verified, which may affect the reproducibility and generalization of the method .

3. Experiments are only conducted in two simulated environments (SimplerEnv and ManiSkill3) and do not involve real physical robotic scenarios, making it impossible to verify the method’s robustness under real-world noise (e.g., sensor errors, robotic arm delays). Furthermore, tasks are concentrated on single-object manipulation (e.g., Pick-and-Place, Push/Pull), lacking validation on more complex long-horizon tasks such as multi-object interaction and dynamic environments, which makes it difficult to fully demonstrate the generalization ability of the IPI pipeline .

**Questions:**

see weakness

---

### Official Review · Reviewer_4Cwm · 2025-10-31

**Soundness:** 3
**Presentation:** 3
**Contribution:** 3
**Rating:** 4
**Confidence:** 5

**Summary:**

This paper proposes a module named STARE (Stage-Aware Reinforcement), designed to address the issues of ambiguous credit assignment and training instability when fine-tuning Vision-Language-Action (VLA) models on long-horizon robotics tasks. The STARE module decomposes complex action trajectories into multiple semantically meaningful stages (e.g., "Reach", "Grasp", "Transport") and provides dense, aligned reinforcement signals for each stage. Building on this module, the authors further develop STA-TPO (for offline stage-wise preference optimization) and STA-PPO (for online intra-stage interaction), and finally integrate them into a three-stage fine-tuning pipeline called IPI (Imitation-Preference-Interaction), which achieves state-of-the-art (SOTA) performance on both the SimplerEnv and ManiSkill3 benchmarks.

**Strengths:**

1. The core insight of this paper is intuitive and powerful: robotic action trajectories possess a strict, causal stage-based structure that differs from linguistic sequences, making stage-wise optimization more effective than end-to-end trajectory optimization.

2. The proposed IPI framework (SFT -> STA-TPO -> STA-PPO) is a logically clear and structurally complete three-stage fine-tuning pipeline that skillfully combines the respective advantages of imitation learning, offline preference optimization, and online reinforcement learning.

3. The experimental results are outstanding, achieving near-perfect success rates (98.0% and 96.4%, respectively) on the two standard simulation platforms, SimplerEnv and ManiSkill3, significantly outperforming existing baselines like SFT, GRAPE (TPO), and RL4

**Weaknesses:**

1. The definition and segmentation of "stages" within the STARE module rely on hand-crafted rules (e.g., based on end-effector position and orientation thresholds), which may limit the method's flexibility and automation when generalizing to more complex and diverse new tasks.

2. The paper's validation is conducted entirely in simulation; despite achieving SOTA results, it lacks experiments on real robots to demonstrate the method's robustness in handling real-world physical noise, latency, and uncertainties.

3. The design of the stage cost functions and potential functions (as shown in Appendices D and E) appears highly task-specific, which raises questions about the engineering complexity required to design these functions for new tasks and the algorithm's sensitivity to their design.

**Questions:**

See weakness

**Details Of Ethics Concerns:**

Nope

---

> ### Author Response · Authors · 2025-11-27
>
> **On the concerns about “limited generalization to more diverse tasks” and “highly task-specific” design.**
>
> We thank the reviewer 4Cwm for bringing up these points. The stage costs in STARE, although illustrated with different task examples in the appendix, are *not* engineered per task. They come from a small, shared set of task-agnostic geometric priors (reach distance, object height change, goal-distance potential...) that stay the same across SimplerEnv and ManiSkill3 tasks. So, despite the variety of tasks, the cost design remains general and not tuned for each case.
>
> As for generalization to more complex tasks—this is not the focus of our work. STARE is focused on a very concrete problem in VLA RL fine-tuning: unstable and inefficient optimization due to unclear credit assignment. That’s the gap we’re addressing. Moreover, for typical tabletop manipulation tasks, the stage definitions we use are intentionally simple and broadly applicable. Our contribution is about offering a new perspective on how to bring stage structure into VLA RL optimization in a **general, stable, and data-efficient** way.
>
> We’d value your thoughts on these clarifications and thank you again for your helpful feedback.
>
> BR,
>
> The Authors

---

### Note · Program_Chairs · 2025-12-15
**Submission Desk Rejected by Program Chairs**

Line 758 violates anonymity